# communications
# engineering

# China's recycling potential of large-scale public transport vehicles and its implications

Xin Xiong[1,2], Xianlai Zeng [1✉], Zhengyang Zhang [3], Robert Pell[4], Kazuyo Matsubae[3] & Zhaoji Hu[2]

Transport infrastructure allows society to function. Such systems continuously improve through manufacturing transformation and technology upgrading. However, its metabolism mechanism of material degradation and quantities from consumption to obsolescence remains unclear. Here we estimate the recycling potential of typical large-scale public transport vehicles (LPTV) in China, in particular, predicting the end-of-life quantity of railway and aviation equipment between 2000 and 2050. Their total recycling potential has been experiencing rapid growth. The total obsolescence mass in 2020 exceeded 33 million tons, and it is expected to reach another 74 million tons by 2050, roughly twice the amount in 2020. By 2050, waste LPTV in China will contain at least 72 million tons of steel, 838 kilotons of aluminum, 2539 tons of titanium, and 223 tons of neodymium. We also compare waste LPTV to e-waste and end-of-life private vehicles. Interestingly, their growth of generation quantity indicates a distinct industry succession from an industrial ecological perspective.

[1] State Key Joint Laboratory of Environment Simulation and Pollution Control, School of Environment, Tsinghua University, Beijing 100084, China. [2] School of Resources & Environment, Nanchang University, Nanchang, Jiangxi 330031, China. [3] Graduate School of Environmental Studies, Tohoku University, Sendai 980-8572, Japan. [4] Minviro Ltd, 25 Lavington Street, London SE1 ONZ, UK. ✉email: xlzeng@tsinghua.edu.cn

Technological progress has produced a variety of products for human needs, which enriches human life[1]. Large-scale public transport vehicle (LPTV) typically consists of railways, aviation, and some associated smart equipment. In recent years, China has formulated a series of plans, policies, and measures to promote LPTV industry. In December 2019, the Ministry of Transport issued guidance on advancing the construction of large databases for improving the efficiency of infrastructure and vehicle systems[2]. In May 2020, a governmental work report of China further pointed out that more efforts should be done to promote the upgrading of high-tech manufacturing. The development of LPTV manufacturing industry is an urgent strategic need to revitalize the entire equipment manufacturing industry and provide equipment and service guarantees for the development of new industries.

In LPTV industry, rail and aviation equipment is the backbone of the modern transport system. The railway equipment mainly includes railway locomotives (RL), railway passenger car (RPC), railway wagons (RW), and high-speed trains (HST). The aviation equipment mainly consists of large and medium aircraft (LMA) and general aviation aircraft (GAA) (see LPTV list in Supplementary Table 1). With the rapid development of LPTV manufacturing and the continuous improvement of transport infrastructure, an increasing number of resources are flowing into this new industry. Accordingly, some metals used and sealed in LPTV have increased dramatically, which may exacerbate the shortage of its geological resources.

While LPTV, like other products, reaches the end-of-life, waste LPTV is generated as the wasteland. Its double properties of environmental risk and resource recycling are calling for a feasible treatment. Waste LPTV contains dozens of valuable metals (e.g., steel, aluminum, titanium, and neodymium) and is known as essential "anthropogenic minerals"[3]. If all waste LPTV are properly and fully recycled, it could help increase metal circularity through a circular economy[4]. This is particularly important for some resource-strapped countries, which must overcome resource shortages and environmental deterioration to achieve sustainable development.

Recycling potential is defined as the recycling amount of generated waste or all the contained materials[5,6]. Estimating the generation amount and recycling potential are the most fundamental prerequisite to harnessing solid waste. Previously, research has concentrated on some conventional end-of-life products, such as waste electrical and electronic equipment (WEEE or called e-waste)[7–10], end-of-life private vehicle (ELPV)[11–14], and waste plastic[15–17]. However, there has been a lack of research and discovery on waste LPTV. As an emerging waste stream, the recycling potential, and its evolution mechanism of LPTV are still unknown. Therefore, this study quantitatively assesses the recycling potential and economic benefits of LPTV, which will be beneficial for promoting resource recycling and ecological civilization construction, and provide guidance for formulating corresponding governance plans, policies, and regulations.

In addition, ecology tells us that community succession exists as the environment or resource changes over time[18]. The growth of one community alters the environment which leads to that community's disappearance and the emergence of a new community which continues to occur in succession. Industrial ecology has a similar phenomenon of succession from emergence, boom, innovation, and substitution of the technical products[19]. The new field called industrial ecology (alimenting from nature ecology) is the study of systemic relationships between society, the economy, and the natural environment. Similarly, the technical product is generally replaced by another high-performance product. The consumer electronics and vehicle industries boomed in the years 2005 and 2010, respectively[20,21]. LPTV manufacturing, starting from around 2015, is at an advanced stage of industrialization[22]. In this work, we will also try to examine the possible succession of technical products from the insight of their waste generation.

Uncovering the generation of waste LPTV could be enabled by appropriate methods and models, such as the basic market supply method[23], the possession coefficient method[24–26], stock-based models[27], market supply A method[28], the Stanford method[29], consumption and use approach[28], material flow analysis[30], and time-series models[31]. The choice of models depends on data availability, reliability, and robustness of models (Supplementary Table 2)[7,32]. In order to estimate the generation of waste LPTV in China, we collected all available data, mainly including the production amount, possession amount, and mass composition of LPTV (Supplementary Tables 3–5), Among them, the future possession amount of LPTV will be predicted by establishing a logistic model; Then we use the possession coefficient method to estimate the generation of waste LPTV, and validate the results using the market supply A method.

## Results

**Waste LPTV generation.** To verify the accuracy of the possession coefficient method, we used the possession coefficient method and the market supply A method to examine the obsolescence amount of railway equipment. The obsolescence amount of RL, RPC, and RW in 2020 is estimated to be 759, 1992, and 39,539 respectively by using the possession coefficient method; The market supply A method is employed to estimate the obsolescence amount of RL, RPC, and RW in 2020 as 664, 2180 and 32,092, respectively (Fig. 1a–c). The obtained results show that

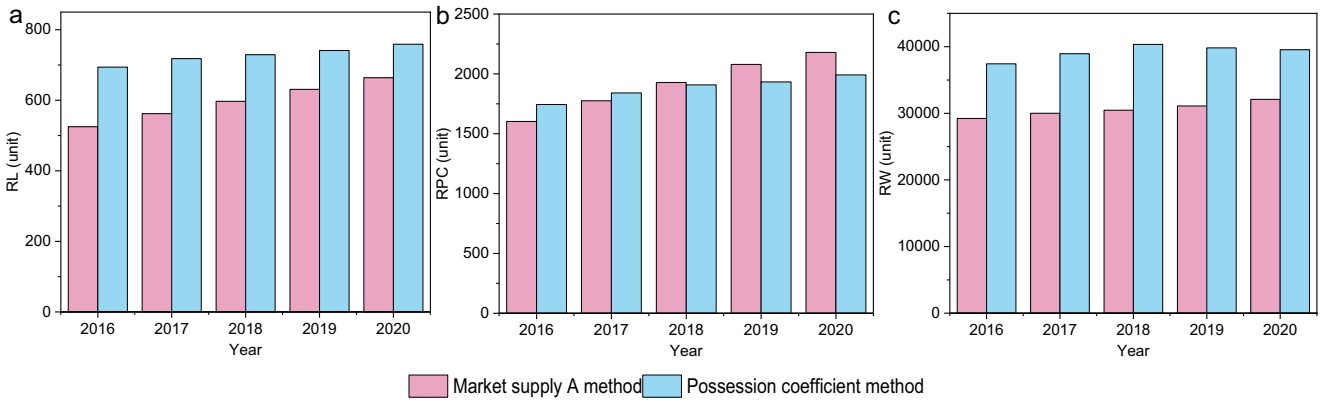

**Fig. 1 Comparison of obsolescence amount using the market supply A method and the possession coefficient method. A** obsolescence amount of RL. **B** obsolescence amount of RPC. **C** obsolescence amount of RW. RL railway locomotives, RPC railway passenger car, RW railway wagon.

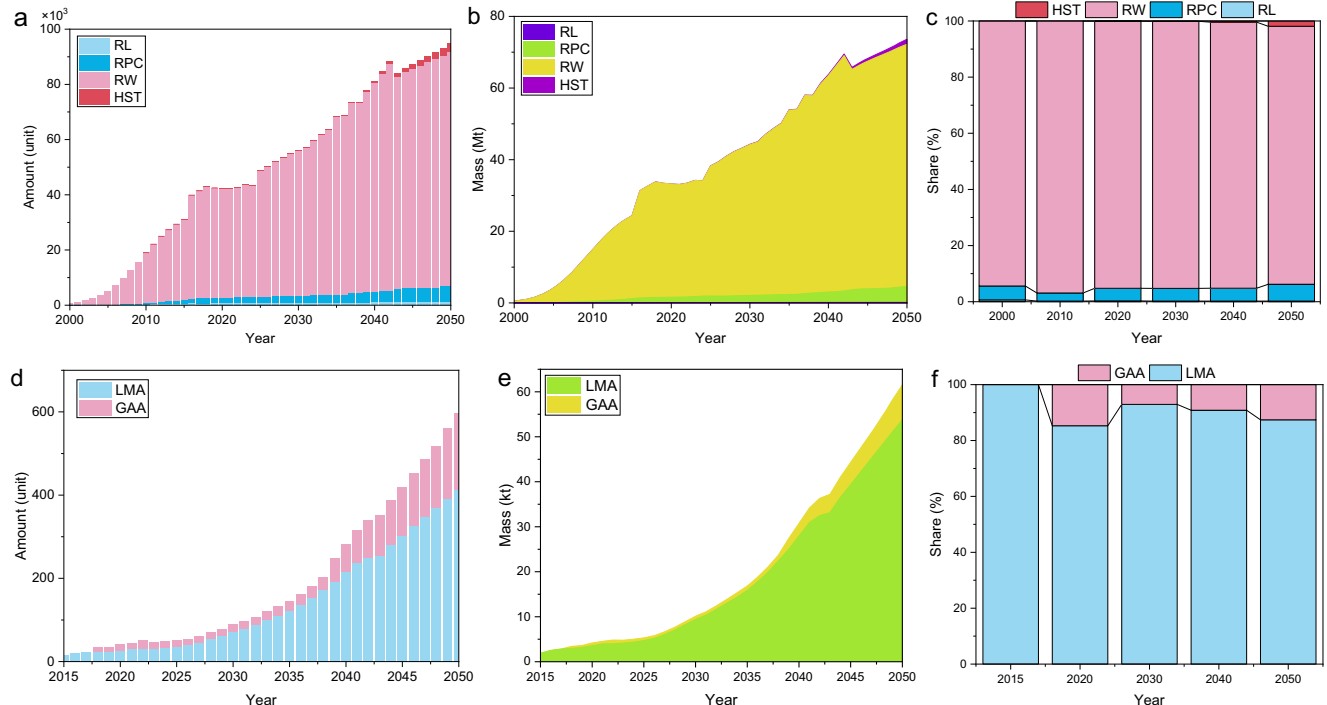

**Fig. 2 Estimation of China's waste LPTV from 2000 to 2050. a** Obsolescence amount of railway equipment. **b** Obsolescence mass of railway equipment. **c** Mass share of obsolescence railway equipment. **d** Obsolescence amount of aviation equipment. **e** Obsolescence mass of aviation equipment. **f** Mass share of obsolescence aviation equipment. RL railway locomotives, RPC railway passenger car, RW railway wagon, LMA large and medium aircraft, GAA general aviation aircraft.

the average difference between the two methods is 15%, with no significant difference. Therefore, the possession coefficient method can feasibly estimate the generation of waste LPTV. The following will use the possession coefficient method to estimate the generation of six types of waste LPTV between 2000 and 2050.

For railway equipment, from 2000 to 2050, its obsolescence amount showed a rapid growth trend (Fig. 2a). In 2000, the total obsolescence mass of railway equipment reached 0.5 Mt (1 Mt = 1000 kt = $10^6$ ton; 1 ton = $10^3$ kg), which will increase tenfold in 2006. After that, it will reach 33.2 Mt in 2020 and 73.6 Mt in 2050 (Fig. 2b and Supplementary Fig. 1). Among them, RW has the largest obsolescence (calculated by mass), accounting for 92.8% ~ 93.7% of the total; The second is RPC and RL, accounting for 4.1% ~ 6% and 0.2% ~ 0.6% of the total, respectively; Finally, the obsolescence mass of HST is the lowest (Fig. 2c).

For aviation equipment, its obsolescence amount will also continue to increase (Fig. 2d and Supplementary Fig. 1). In 2015, the average obsolescence mass of aviation equipment was 1.9 kt, but it will reach 4.2 kt in 2020 and 61.5 kt in 2050 (Fig. 2e). From 2015 to 2050, the average scrap growth of aviation equipment will reach 1.7 kt per year. The contribution ratio of LMA to GAA is expected to remain approximately 9:1 (Fig. 2f).

Totally, in 2000, the obsolescence mass of LPTV was 0.5 Mt, but it will reach 33.2 Mt by 2030 and 73.7 Mt by 2050, with an annual growth rate of 10.6% (Supplementary Fig. 1g). Among them, railway equipment is the main contributor of waste LPTV, and its obsolescence accounts for more than 99% of the total mass. In addition, economic growth significantly accelerates the generation of waste LPTV[33]. The regression model between per capita gross domestic product (GDP) and scrap per capita of LPTV is shown in Fig. 3. From 2010 to 2020, the scrap per capita increased by 12 kg, which showed that as the economy grew, demand for LPTV increased, which led to higher waste flows from this category.

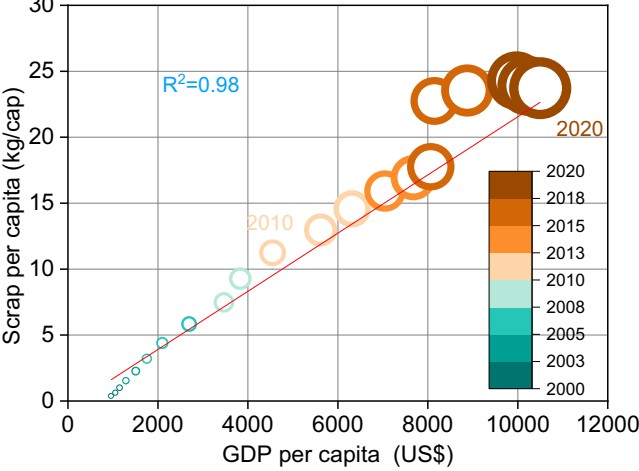

**Fig. 3 Relationship between per capita GDP and scrap per capita.** The continuous enlargement of the circle represents the increase of the year. Data source from Supplementary Tables 8 and 9.

As the future generation mass of waste LPTV is determined from the forecasted possession amount, it is necessary to evaluate the possession amount forecast data scientifically. This study will take railway equipment as an example to compare and discuss the railway equipment quantity with that of some developed countries. China has set the goal of developing into a transportation power and a moderately developed country by 2050, which means that the possession amount of LPTV in China may reach a peak around 2050. The obtained data indicate that in recent years, the possession amounts of railway equipment in developed countries such as Japan, Germany, and the United Kingdom has maintained as a peak (Supplementary Table 6). Here, this study

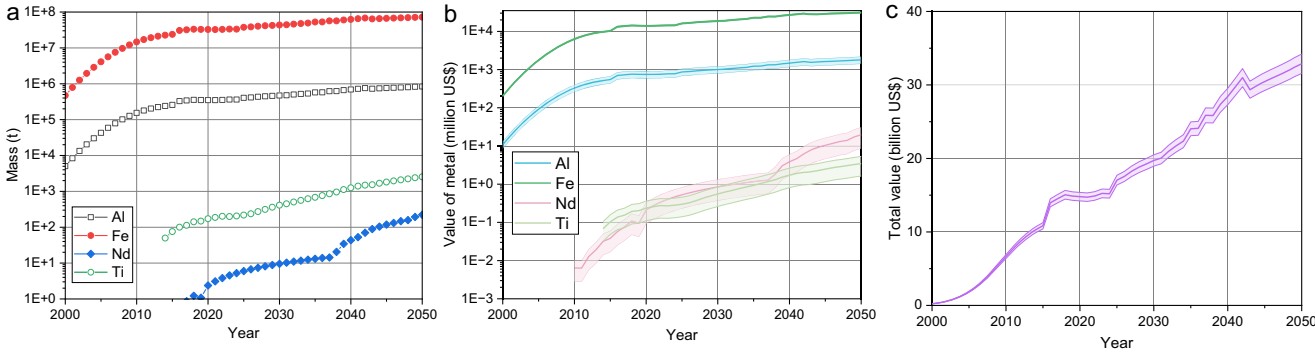

**Fig. 4 Projected recycling potential of typical metals in LPTV. a** Mass of Al, Fe, Nd, and Ti. **b** Economic potential of typical metals. **c** Total economic recycling potential. The shaded area indicates the value range.

will compare the predicted per capita possession data of China in 2050 with the per capita possession data of the above three countries in 2019. It is found that the per capita possession of RL, RPC, RW, and HST in these three countries is about 0.1 kg ~ 6 kg, 137.9 kg ~ 161.5 kg, 99.3 kg ~ 765.7 kg and 56.4 kg ~ 92 kg. The per capita possession of RL, RPC, RW, and HST in China is 2.5 kg, 91.6 kg, 635.3 kg, and 65.5 kg (Supplementary Table 7), basically close to the level of these three developed countries.

**Recycling potential and environmental benefit**. When LPTV reaches the EoL stage, whether it is in hibernation or not, it will inevitably contain a large number of valuable resources[34]. Fe, Al, Ti, and Nd, as the typical metals in the waste LPTV, maintain a growing trend in their content. In 2020, the encapsulated metals of Fe, Al, Ti, and Nd are about 32.5 Mt, 350.5 kt, 172.9 t, and 2.4 t, respectively, but by 2050, they will rise to 71.9Mt, 838.2 kt, 2538.7t, and 223.1 t, respectively (Fig. 4a). Due to the rapid development of HST, the growth rate of Nd will continue to accelerate. From an economic point of view, the four typical metals accumulated in waste LPTV also enhance the recovery potential (Fig. 4b). In 2000, the average economic potential of the four metals was 0.2 billion, but it will reach 14.8 billion in 2020 and 32.9 billion in 2050. Among them, Fe and Al totally account for about 99% of the economy shares (Fig. 4c). Therefore, Fe and Al will be the leading recycling targets in LPTV.

In the increasingly serious situation of global warming, carbon emission reduction has become a consensus formed by the major powers. Waste LPTV contains plenty of metals, whose primary mining process could generate a large amount of $CO_2$ emission. In contrast, urban mining through material recycling largely produces much fewer emissions (Supplementary Fig. 2)[35,36]. Therefore, the recycling of LPTV is particularly meaningful to reduce emissions. In this study, the carbon footprint of four typical metals in raw and secondary material production was collected to assess the environmental benefits of recycling[37] (Supplementary Table 10). If these typical metals are properly recovered, Fe, Al, Ti, and Nd can approximately reduce carbon emissions of 76.2 Mt, 4.4 Mt, 2.1 kt, and 1.6 kt, respectively by 2050. The carbon reduction benefits of the four metals will reach 80.6 Mt on average (Supplementary Fig. 3).

**Sensitivity and uncertainty analysis**. Due to the possible product substitution, the service life of LPTV may change, so sensitivity analysis must be carried out for the change of service life. The study found that when estimating the generation of waste LPTV in 2010, one year of life extension can reduce the generation of waste by 16.8%, and one year of life reduction can increase the generation of waste by 17.4%. However, there will be no deviation of more than 4.3% in the output after 2015 (Fig. 5a). This shows

that the impact of short-term estimation is significant, but in the long run, the production of waste LPTV will not be excessively affected by the service life of LPTV, which verifies the robustness of the estimation of the production of waste LPTV in this study.

Based upon the collected data (Supplementary Tables 5, 11), uncertainties of waste LPTV mass and the total stock of Fe and Nd are performed and presented in Fig. 5. Waste LPTV mass in 2020 and 2050 in Supplementary Fig. 1g can be excellently reflected by the simulated distribution, given in Fig. 5b and Fig. 5c. Similarly, Monte Carlo simulation for metals (like Fe and Nd) stock at the maximum probability interval can also cover the forecasting results (Fig. 4a and Fig. 5d, e). All the uncertainty analysis can substantially verify and validate the accuracy and robustness of the above estimation and findings.

## Discussion

EEE, private vehicles, and LPTV are all high-tech industries. Here, we try to use these three types of industry and thirty-seven categories of product wastes to uncover the industry succession (Fig. 6). The increase in product waste generation can reflect the rise, prosperity, and decline of an industry. Regarding WEEE, all the types of WEEE in China kept a high increasing rate before 2015. The majority will gradually decrease to near zero increasing the rate until 2050. DC, camera, FM, and SMT will demonstrate a faster rate of decline. Negative growth in 2050 suggests that industries such as DC, FM, cameras, and SMT are shrinking and being replaced by other products. DC and cameras, for example, have been partly replaced by smartphones with advanced photography functions. At the end of the 20th century and the beginning of the 21st century, with the rapid development of science and technology, new technology and electronics quickly enter the market, and old equipment is gradually replaced by new ones, resulting in a rapid increase in the amount of e-waste. From 2011 to 2025, however, the growth rate of e-waste declined from 23.7% to 5.6% indicating a potential substitute for technical products and further implies an ongoing industrial succession.

The vehicle also demonstrates a similar evolution to EEE. Most of ELPVs like passenger cars would maintain a significant growth by 2025 and EVs will continuously rise at least until 2040[38]. In addition, by virtue of China's vigorous implementation of the subsidy policy for the purchase of EVs, the improvement of charging stations, and the optimization of road driving, the number of electric vehicles has shown a continuously explosive growth trend in recent years[39]. In 2020, the obsolescence growth rate of EV has reached 100%. Conventional vehicles like PV and bicycles would reach zero or a negative increasing rate, which was partly replaced by EV and e-bicycle. Among the six types of LPTV, their main increasing rates in 2010–2050 would be greater than zero. RPC, RW, and RL had a distinct increasing rate before

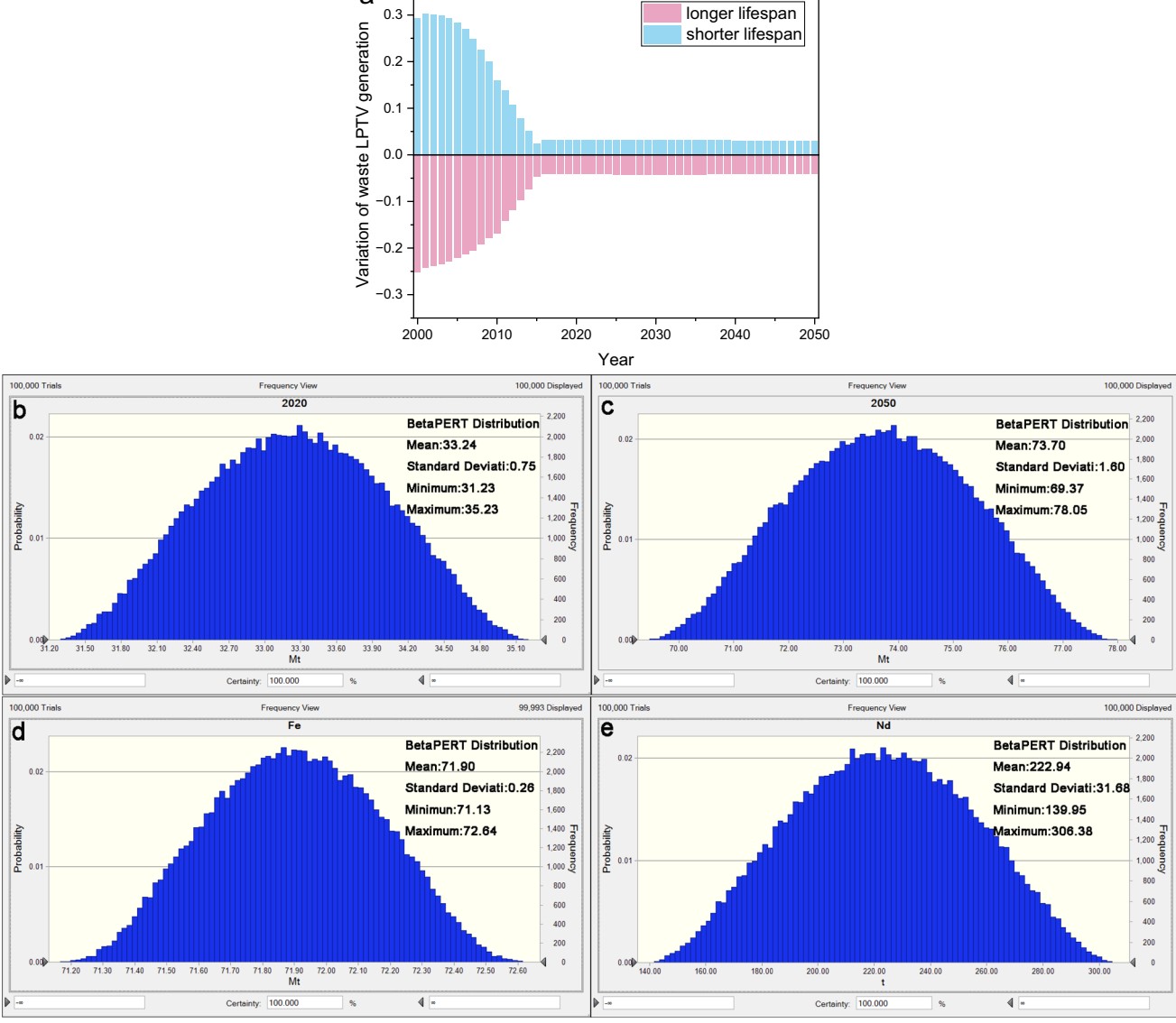

**Fig. 5 Sensitivity and uncertainty analysis. a** Mass of waste LPTV with 1-year variation. **b** Total mass of waste LPTV in 2020 affected by each mass. **c** Total mass of waste LPTV in 2050 affected by each mass. **d** Fe mass of yearly-generated waste LPTV in 2050 affected by resources content. **e** Nd mass of yearly-generated waste LPTV in 2050 affected by resources content.

2015 and afterward would fall to zero by 2050. In contrast, HST, LMA, and GAA grew significantly (by over 40%) between 2010 and 2050. They will dominate as leading LPTV industries in three decades (Fig. 6). Finally, there are some industrial successions: in the EEE industry, cameras, DC, and TV are partly being substituted by mobile phones; PV, car, and bicycles are being replaced by electric transportation the carbon emission reduction; the LPTV industry will totally boom by 2050, especially for HST, LMA, and GAA.

Waste LPTV's growth rate is generally declining, but the obsolescence is increasing. First, due to the rapid development of the economy, people's travel needs are constantly increasing, and more LPTVs are needed to meet their travel needs. Secondly, China has a large population and a high demand for LPTV. When LPTV reaches the end-of-life stage, it will generate a huge amount of waste LPTV. Thirdly, with the gradual improvement of transportation construction, the number of LPTV needed will be saturated, and the growth rate will also decline.

According to the prediction results, waste LPTV capsulated a large mass and rich materials (Fig. 4). If these resources are

recycled, they can reduce energy consumption and pollutant emissions to a certain extent, bring significant economic benefits to the relevant recycling industry, and promote the healthy development of environmental protection and LPTV industry[14]. However, compared with WEEE and ELPV, LPTV recycling is still in its infancy. Firstly, due to the large volume, very complex composition, and toxic materials of LPTV, LPTV dismantling is complicated, and parts are easily lost. The current level of dismantling is still difficult to meet the needs. Secondly, the number of enterprises with dismantling capacity in China is very limited. Taking railway equipment as an example, China established its first formal dismantling company of trains in 2021, with a dismantling capacity of 4000 vehicles per year, which is far from satisfying the number of waste trains. Finally, the scrap recovery management system of LPTV has not been improved and there is no corresponding circular economy policy[40]. Therefore, more innovative waste LPTV recycling technologies should be supported so that the overall waste LPTV recycling efficiency can be increased. In this regard, university-industrial cooperation is key so that the key researchers can better understand the real needs

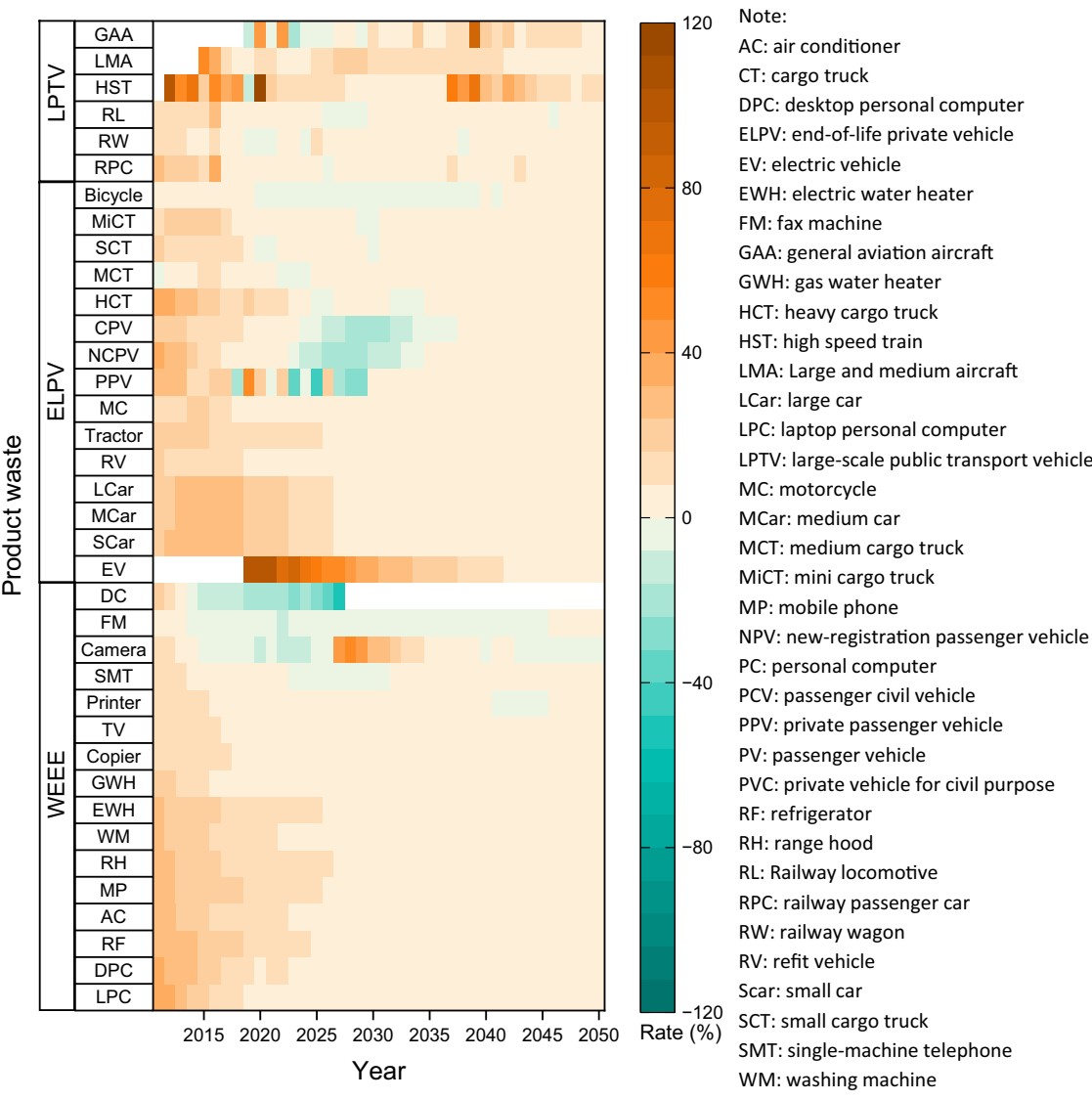

**Fig. 6 Growth rate of product waste generation in 2010-2050 in China.** Data source from Supplementary Data 1–4 and the abbreviations and acronyms of main vocabulary from Supplementary Note 1.

from their industrial counterparts and the proposed technologies can be tested and applied promptly. The focus should be on railway equipment, as railway equipment is the main contributor of waste LPTV. Moreover, a national waste LPTV resource information platform should be created to share the relevant data and information among different stakeholders. Such a platform can also facilitate domestic technological cooperation so that advanced technologies can be quickly transferred and applied. This study can provide science-based references for governments and relevant departments to formulate plans and policies concerning anthropogenic circularity and can be of great significance to the sustainable development of the LPTV industry.

## Methods

**Market supply A method**. The market supply A method[28], also termed as "distribution delay method" in some literatures[41], refers to a method that estimates the obsolescence amount based on the sales amount and lifespan distribution. The basic equation to estimate the generation of waste LPTV is shown in Eq. 1[24].

$$Q_z = \sum S(x) \cdot f(x)(1 \le x \le l) \tag{1}$$

where $Q_z$ represents the obsolescence amount of $z$ years; $S(x)$

represents the previous sales amount for $x$ years; $f(x)$ represents the obsolescence ratio at $z$ year(s) of such equipment manufactured $x$ years ago; $z$ represents the longest service life of the equipment. The *China Railway Yearbook* presents detailed and high-quality production data of trains, so the sales amount in this work was replaced by the production amount for calculation.

The obsolescence ratio of $f(x)$ can be obtained by Weibull distribution function[42]. This function is widely regarded that can effectively reflect the distribution of the service life density of LPTV, with the advantages of good fitting data and convenient processing of equations. The probability density function and cumulative distribution function are presented in Eq. 2 and Eq. 3 (Supplementary Fig. 4)[43,44]:

$$f(x) = \begin{cases} \frac{\beta}{\eta}\left(\frac{x}{\eta}\right)^{\beta-1} e^{-(x/\eta)^{\beta}}, & x \ge 0 \\ 0, & x < 0 \end{cases} \tag{2}$$

$$F(x) = 1 - e^{-(x/\eta)^{\beta}} \tag{3}$$

where $\beta$ refers to shape parameter, and $\eta$ refers to scale parameter (Supplementary Table 12).

**Possession coefficient method**. The possession coefficient method estimates the obsolescence amount based on the possession amount, as shown by Eq. 4. Assuming that the possession amounts of LPTV in a given year will reach the peak of obsolescence in the next (a, b) years, then the obsolescence amount can be obtained by the dynamic stock models[45–47]:

$$Q_{t+b} = \frac{C \cdot P(t)}{b - a + 1} \tag{4}$$

where $Q_{t+b}$ represents the obsolescence amount of $t + b$ years; $P(t)$ represents the possession amount for $t$ years, and $C$ is the cumulative obsolescence ratio, corresponding to different peak times of obsolescence. It is usually set as 50% ~ 70%. Based on Weibull distribution, $C$ is set as 60% in this study. $(a, b)$ represents the interval of peak obsolescence years (Supplementary Table 13).

**Data regression**. We use the time-step method based on data regression to predict the future possession amounts of LPTV. The logistic function was put forward by the mathematician Verhulst in the process of studying the problem of population growth. In recent years, the logistic model has been widely used to predict the generation of WEEE and ELPV. The growth of the possession amount of product can be divided into four stages: primary introduction, rapid growth and saturation, then final decline, its growth curve is "S" shape[48]. In addition, China issued a policy in 2021, that is, China will accelerate the construction of a transportation power in the next three decades, basically build a transportation power by 2035, and fully build a transportation power by 2050. This means that the number of LPTV may reach saturation in 2050. Based upon the above modeling principles and policies, this study fitted the logistic curve according to the past possession data (Supplementary Table 4 and Supplementary Fig. 5), and the future possession amount of LPTV can be calculated from Eq. 5:

$$P(t) = \frac{p_{max}}{1 + ae^{-b(t-t_0)}} \tag{5}$$

where $P(t)$ represents LPTV possession amounts in year $t$; $t_0$ is the first year in the period; $p_{max}$ represents the saturation value of LPTV possession amount; and $a$ and $b$ are two parameters describing the growth rate.

**The metal resources in waste LPTV**. When dealing with the waste LPTV, it is very important for stakeholders to determine the recycling priority of materials[49]. Considering factors like market value, environmental impact, and the scarcity of resources, some scholars proposed concepts of "metal criticality" and "recyclability" to recycle selected target materials. The economic values of recycled resources in LPTV were assessed in this study, and the generation mass and market value of metal resources can be determined by Eq. 6 and Eq. 7, respectively[14]:

$$V_j = \sum_{i=1}^{n} W_i(z) \cdot e_{ij} \tag{6}$$

$$T_{avg} = \sum_j p_{j(avg)} \cdot V_j \tag{7}$$

where $V_j$ is the mass of metal resource $j$ within $n$ categories of LPTV; $i$ is the $i$th category of LPTV; $W_i(z)$ is the total mass of the $i$th category of LPTV that is wasted in the year $z$; $e_{ij}$ is the proportion of metal resource $j$ contained in $i$th category of LPTV (Supplementary Table 11). $T_{avg}$ is the total average market value of all metal resources, and $p_{j(avg)}$ is the market price per ton of metal resource $j$ within a given period (Supplementary Table 14). As the proportions of metal resources in different types of LPTV

were not determined, the average resource content was thus adopted as an appropriate value. In addition, assuming that the metal resources in LPTV would not lose during service life, then the metal resource losses can be negligible in the process of LPTV approaching its end of service life[14].

**Sensitivity and uncertainty analysis**. In order to evaluate the accuracy of the results obtained, sensitivity and uncertainty analysis is usually used to verify the results. Specifically, the most common method for sensitivity analysis is to modify one input parameter at a time and observe the impact of changes in model input on model output. In addition, uncertainty analysis quantifies the imprecision in model predictions and investigates the variation of model outputs in response to given reasonable distribution of data inputs[8,50,51].

This study considers that the life distribution is the main factor affecting the prediction result of the production of LPTV. Therefore, sensitivity analysis will be carried out on the life of LPTV. The data including mass and average content will be assessed with uncertainty analysis. Here, a Monte Carlo simulation ($10^5$ iterations) was conducted to obtain final estimates of flows and their uncertainties in this study (Supplementary Fig. 6).

**Reporting summary**. Further information on research design is available in the Nature Portfolio Reporting Summary linked to this article.

## Data availability

The data that support the findings of this study are available from the corresponding author upon reasonable request. A detailed description of methods; equations; data sources; glossary; and other informing results: Supplementary Tables 1–14, Supplementary Figures 1–6, Supplementary Note 1, and Supplementary Data 1–4.

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

## Acknowledgements

The work is financially supported by Tsinghua University- Tohoku University Collaborative Research Fund 2021, the National Natural Sciences Foundation of China (92062111), and the National Key Technology R&D Program of China (2019YFC1908501). We gratefully acknowledge Prof. Daqian Jiang and Dr. Guochang Xu for their valuable comments.

## Author contributions

Conceptualization, funding acquisition: X.Z. Methodology: X.Z. and K.M. Investigation, resources: X.X. and X.Z. Writing – Original Draft: X.X. Writing –Review & Editing: X.Z. and R.P. Supervision: X.Z., K.M., and Z.H.

## Competing interests

The authors declare no competing interests. Xianlai Zeng is acting as Guest Editor for a collection in *Communications Engineering*, but was not involved in the editorial review of, nor the decision to publish this article.
