## [Peer Review File · Communications Engineering]

Reviewers' comments:

Reviewer #1 (Remarks to the Author):

Overview:

The article puts a focus on rising waste flows from high-end equipment, with China as a case study. The authors built scenarios for future waste generation amounts from this broad product category, which includes railway and aviation equipment, and aircrafts.

Strong points of this work include the relevance and novelty of the topic, the comprehensive results, and the uncertainty analysis.

There are a number of major weaknesses of the current manuscript, however, which severely lower its quality. The issues that I raise are too many and too severe for this manuscript to become part of the literature in its current form. The quality of the manuscript is clearly below the average quality of manuscripts from Chinese authors, of which I review around five each year. Therefore, I recommend rejection to the editor.

The issues I raise can in principle be fixed and the manuscript improved, which is why I also recommend a resubmission after substantial revision.

Major comments:

The manuscript does not contain line numbers. There are a number of incorrect formulations (Like second paragraph "The railway equipment mainly includes..." should be "The aviation equipment mainly includes...", but given that line numbers are absent, I will not report these here.

The abstract needs to adopt a more scientific language at several places. First sentence: "High-end equipment is a powerful new engine for manufacturing transformation and technology upgrading." is not scientific writing "powerful new engine..." and should be rephrased to contain examples and describe exactly which technologies can be upgraded how. The same holds for the phrase "quantities from wonderland to wasteland", which is not descriptive enough for a scientific text. The final sentence "indicates a distinct industry succession mechanism from an industrial ecological perspective" must also be written in a more understandable fashion. What industries and what is the succession here?

The category "high end equipment" is not properly defined. Nowhere in this work. From the introduction, the reader gets the impression that electric vehicles (ELV) are not included as they form a separate ELV category and that only railway and aviation equipment are included (e.g., Fig. 2). In figure 6, however, vehicles and appliances are also listed. Please add a list of products that you consider high end equipment to the work, for example, at the start of the supplementary material.

Between the introduction and the results section, a short overview of the results and method paragraph must be introduced (see journal guidelines). Right now, the break between introduction and results is too abrupt. The results start with the use of "market supply A method", without further explanation, but the authors should at least state what are the main modelling principles, data, and time frame etc. It

should also be stated that these methods had to be applied because actual waste statistics were not available (which I assume is the case). Please clarify!

“Since HEE is an emerging industry, its future production perhaps maintains continuous linear growth. Linear fitting was employed to quantify the ownership of six categories of HEE from 2021 to 2050 (Figure S3),” That is a very strong assumption and requires justification. At least, you should compare the resulting per capita ownership values with large countries/regions with a higher GDP per capita such as the US or the EU. A better justification of this assumption is needed. Also, Figure S3 shows something else but I could find the future ownership rates in Table S4.

“From the viewpoint of recycling, these four types of products suffer from some difficulties in recycling and a large quantity of treatment.” Such a sentence (and there are many of this kind in the text) is not acceptable for a scientific text. First, no reference is given. Second, “some difficulties” is not specific or descriptive and far too vague. The reader who shall spend time on reading your manuscript deserves more detailed and exact information. Moreover, this sentence should be part of discussion and not of the results section. Below, you refer to “large weight amounts” and “low recyclability”, but until 2050, recycling capacities can be built up and better recycling technologies be developed, especially in China, where innovation and industry development are very quick!

Title, Figure 2, and text. The indicator “recycling potential” is not defined and not explained. Is it the total mass of all waste equipment or just the fraction that is potentially recyclable and if so, with what technology? This is crucial information and needs to enter the main text, not just the appendix.

Fig. 5 is not clear to me. The red and green shaded areas are not explained (please change color coding to accommodate color-blind readers!), the yellow curve does not show in the pdf version of the manuscript. It looks like that the plot in the pdf is not displayed correctly. Please check and correct.

Fig. 6: It is not clear which rate is shown here. Growth rates (%) or flow rates (Mt/yr?)

The entire manuscript suffers from a constant mix of introduction, results, and discussion elements. This makes it hard for the reader to follow. For example, in the discussion, suddenly the following phrase appears: “China is the world's largest producer of e-waste 30. The growth rate of e-waste will decrease from 23.7% to 5.6% from 2011 to 2025.” That clearly belongs to the introduction. Please make a larger effort to disentangle the different parts of the manuscript to generate a more logical flow of the text.

The methods section still contains text from the manuscript guideline and the start of the actual content suggests that text by the authors has accidentally been deleted or overwritten.

Minor comments:

Title: Since the case study is on China, this must be reflected in the title. Please change.

Introduction, last paragraph: Since materials are kind of the nutrients of the technical system, it is a good idea to think about ecological succession under the industrial ecology metaphor. However, this has hardly been done in the literature and most readers are not familiar with the concept. Therefore, I think that you must introduce this topic with a language that also non-ecologists understand. In particular, the sentence "In this research, we will also try to examine the succession of technical products from their waste generation." needs rephrasing. What do you mean by "succession of technical products from their waste generation"?

The results shown in Fig. 1 are redundant, as part C shows the same data as parts B and C, just as different graph. Instead, parts A and B should be placed next to each other, that facilitates visual comparison.

In Fig. 3B, the meaning of the size of the circles is not explained. In the related text you write "From 2010 to 2020, the per capita obsolescence weight increased by 9 kg, which showed that China's transportation facilities were gradually improved and more HEE was put into use with the economic growth, thereby increasing per capita obsolescence quantity too." This sentence is hard to understand, please rephrase! ("as the economy grew, demand for HEE increased, which led to higher waste flows from this category...")

Reviewer #2 (Remarks to the Author):

The authors presented research work on the prediction of HEE in the future. The topic is of significance for resource recycling and the corresponding policy-making. Generally, the manuscript is well organized. For improving the quality, the following issues may need more clarity of discussion. Specific comments are listed below.

1. In the model, it was assumed that the production of HEE increased linearly, and based on this assumption, the futural ownership of HEE over the period 2021-2050 was predicted. Considering the long period, such a simplified linear model may induce significant prediction errors. The authors are suggested to comment or discuss this in more detail on this issue.
2. Figure 6 shows the increasing rate of various resources in the next 30 years. It was observed that each kind of resource varies significantly in their increasing rate. It is widely accepted that the increasing rate decreases gradually. However, Fig. 6 shows that some resources show alternating changes of positive and negative increasing rates. It is an interesting point, but more discussion is required to disclose the mechanism of such variation.
3. In the section "Analysis of data error and uncertainties," It is suggested to add a comment on the estimation results concerning the data error and uncertainties. Currently, the authors only described that they had performed the data checking.

Reviewer #3 (Remarks to the Author):

The manuscript combines solid analysis with poor writing. It requires substantial revisions, and the authors are strongly encouraged to seek help with their English. Problems are mostly related to diction. Apart from the writing, there are some questions I have about the modelling.

1. 'High-end equipment'. This term is used to describe railway and aviation equipment. It is entirely misleading, and the rationale for using it seems to be the similarity of the acronym to electric and electronic equipment (EEE). EEE is descriptive of the kind, while HEE is not. Many pieces of EEE claim to be high-end (as a simple internet search for 'high-end equipment' shows), but they are not included in the author's definition. Please use a descriptive name.
2. RL, RPC, RW, HST etc. Nobody can keep track of all the acronyms that the authors needlessly introduce. They adversely impact the readability of the manuscript. Please write out the names!
3. Individual word usage is often problematic. 'Increasing rate' is no term. It is rate of increase or growth rate. It needs to be specified in terms of a time step, for example as a rate per year, at least once in a particular in a figure. Figure 6 cannot be understood without this information. I do not quite understand how the rates can be so high.
4. I have some problems understanding the referencing. Individual claims or statements are not well references. For example, the paragraph of the introduction 'Limited researches ..' cites four references after four sentences, but it is not clear which of the prior statements these references refer to. Since methods are cited, I would recommend providing a reference for each method. I also question why none of the references describing the research methods in the introduction are repeated in the methods section.
5. Fig. 1A is not referred to in the text. There is no explanation of why two different methods were used, in Fig 1A and Fig 1B, to calculate the same quantities, or how to interpret the differences. Presumably, they provide different empirical basis to estimate a quantity that is not observed and hence constitute some form of triangulation.
6. The methods are poorly described. I cannot follow. I do not see that the original method development is appropriately acknowledged. Eq. 1 operates with two different time variables, where one seems to be relative to the other. I have trouble interpreting eq. 4 or understanding what the terms mean and where the data comes from. Please note that cohort or dynamic stock models have been used many years in industrial ecology but have a specific history, which it would be good to acknowledge here.
7. Kg measures mass; weight is measured by Newton. (In imperial units, the pound is colloquially used for both, hence the confusion in English.)
8. Displays starting on p. S15 are not legible. Generally, the SI is difficult to read. References are lacking on S2, and so is a description of what is represented in the different parts.

Detailed Response to Reviewers' Comments

Title: From wonderland to wasteland: recycling potential of typical high-end equipment and its implications

Manuscript ID: COMMSENG-22-0221

Reviewer #1

1. Comment:

The article puts a focus on rising waste flows from high-end equipment, with China as a case study. The authors built scenarios for future waste generation amounts from this broad product category, which includes railway and aviation equipment, and aircrafts.

Strong points of this work include the relevance and novelty of the topic, the comprehensive results, and the uncertainty analysis.

There are a number of major weaknesses of the current manuscript, however, which severely lower its quality. The issues that I raise are too many and too severe for this manuscript to become part of the literature in its current form. The issues I raise can in principle be fixed and the manuscript improved, which is why I also recommend a resubmission after substantial revision.

Response:

Thank you for your positive comments and the problems pointed out. We have made substantial amendments to the manuscript.

2. Comment:

The manuscript does not contain line numbers. There are a number of incorrect formulations (Like second paragraph “The railway equipment mainly includes...” should be “The aviation equipment mainly includes...”, but given that line numbers are absent, I will not report these here.

Response:

Thanks for your comment. We added line numbers to the paper and corrected the errors.

Please check up the tracked-changing manuscript. Thanks !

Remark:

The railway equipment mainly includes railway locomotives (RL), railway passenger car (RPC), railway wagon (RW) and high-speed trains (HST). The aviation equipment mainly includes large and medium aircraft (LMA) and general aviation aircraft (GAA).

3. Comment:

The abstract needs to adopt a more scientific language at several places. First sentence: “High-end equipment is a powerful new engine for manufacturing transformation and technology upgrading.” is not scientific writing “powerful new engine...” and should be rephrased to contain examples and describe exactly which technologies can be upgraded how. The same holds for the phrase “quantities from wonderland to wasteland”, which is not descriptive enough for a scientific text. The final sentence “indicates a distinct industry succession mechanism from an industrial ecological perspective” must also be written in a more understandable fashion. What industries and what is the succession here?

Response: thanks for your v suggestions. The abstract has been rewritten for better understanding in a scientific way.

Remark:

High-end equipment is functioning our society through manufacturing transformation and technology upgrading. However, its metabolism mechanism and quantities from consumption to obsolescence remain unclear. Here we estimate the recycling potential of typical high-end equipment in China and predict the end-of-life quantity of railway and aviation transportation equipment between 2000 and 2050. The obtained results show that their total recycling potential has experienced a rapid growth, exceeding 33 million tons in 2020. It is also expected to reach 74 million tons in 2050, roughly twice the amount in 2020. By 2050, waste high-end equipment in China will contain at least 72 million tons of steel, 838 tons of aluminum, 2,539 tons of titanium and 223 tons of neodymium. We also compare waste high-end equipment to other product waste such as e-waste and end-of-life vehicle. Interestingly, their growth of generation quantity indicates a distinct industry succession from an industrial ecological perspective.

4. Comment:

The category “high end equipment” is not properly defined. Nowhere in this work. From the introduction, the reader gets the impression that electric vehicles (ELV) are not included as they form a separate ELV category and that only railway and aviation equipment are included (e.g., Fig. 2). In figure 6, however, vehicles and appliances are also listed. Please add a list of products that you consider high end equipment to the work, for example, at the start of the supplementary material.

Response:

Thanks for your advice. We added the regulated product list of high-end equipment in the supplementary material. Please check up the supplementary material. Thanks !

Remark:

Supplementary Table1 Categories of typical high-end equipment

Typical high-end equipment	Categories
Railway equipment	Railway locomotives (RL)
	Railway passenger car (RPC)
	Railway wagon (RW)
	High-speed trains (HST)
Aviation equipment	Large and medium aircraft (LMA)
	General aviation aircraft (GAA)

5. Comment:

Between the introduction and the results section, a short overview of the results and method paragraph must be introduced (see journal guidelines). Right now, the break between introduction and results is too abrupt. The results start with the use of “market supply A method”, without further explanation, but the authors should at least state what are the main modelling principles, data, and time frame etc. It should also be stated that these methods had to be applied because actual waste statistics were not available (which I assume is the case). Please clarify!

Response:

Thanks for your comment. We have added modeling principles, data and time range in this section. Please check up the tracked-changing manuscript. Thanks.

Remark:

In order to uncover the generation of China's WHEE, the following four steps will be employed. Firstly,

the spatial boundary of the whole study is the mainland China, and the temporal boundary is the year of 2000-2050. All the available data is collected and pre-mined, mainly including the production, possession, and mass composition of HEE (Supplementary Tables 3, 4, and 5); Secondly, we verify the accuracy of the estimation method of the WHEE production. Thirdly, a logistic model was established to predict the possession amount of HEE, and use the possession coefficient method to estimate the generation amount of WHEE by 2050; Finally, the sensitivity and uncertainty of the obtained results are analyzed to validate the main results.

6. Comment:

“Since HEE is an emerging industry, its future production perhaps maintains continuous linear growth. Linear fitting was employed to quantify the ownership of six categories of HEE from 2021 to 2050 (Figure S3),” That is a very strong assumption and requires justification. At least, you should compare the resulting per capita ownership values with large countries/regions with a higher GDP per capita such as the US or the EU. A better justification of this assumption is needed. Also, Figure S3 shows something else but I could find the future ownership rates in Table S4.

Response:

Thanks for your comment. According to the relevant national policies and historical ownership, we built a logistic model to predict the future ownership. Because the logistic model is more scientific, it has been compared and discussed with the per capita ownership values of developed countries. Figure S3 is quoted incorrectly in the text. We have changed it in the manuscript. Please check up the tracked-changing manuscript. Thanks !

Remark:

Data regression. This study uses the time-step method based on data regression to predict the future possession amounts of HEE. The logistic function was put forward by the mathematician Verhulst in the process of studying the problem of population growth. In recent years, the logistic model has been widely used to predict the generation of WEEE and ELV. The growth of the possession amount of product can be divided into four stages: primary introduction, rapid growth and saturation, then final decline, its growth curve is "S" shape. In addition, China issued a policy in 2021, that is, China will accelerate the construction of a transportation power in the next 30 years, basically build a transportation power by 2035, and fully build a transportation power by 2050. This means that the

number of HEE may reach saturation in 2050. Based on the above modeling principles and policies, this study fitted the logistic curve according to the past possession data (Supplementary Table 4 and Supplementary Fig. 5), and the future possession data of HEE can be calculated from Eq. 5:

$$P(t) = \frac{p_{max}}{1 + ae^{-b(t-t_0)}} \quad (5)$$

where $P(t)$ represents HEE possession amounts in year t ; p_{max} represents the saturation value of HEE possession amount; and a and b are two parameters describing the growth rate. The simulations were performed by Origin.

As the future generation mass of WHEE is calculated on the basis of the possession amount forecast data, it is necessary to evaluate the possession amount forecast data scientifically. This study will take railway equipment as an example to compare and discuss the railway equipment quantity with that of some developed countries. China has set the goal of developing into a transportation power and a medium developed country by 2050, which means that the possession amount of HEE in China may reach saturation in 2050; At present, railway facilities in some developed countries have been very perfect. Research data show that in recent years, the possession amounts of railway equipment in developed countries such as Japan, Germany and the United Kingdom has changed very little and is basically saturated (Supplementary Table 6). Therefore, this study will compare the predicted per capita possession data of China in 2050 with the per capita possession data of the above three countries in 2019. It is found that the per capita possession of railway locomotives, railway passenger car, railway wagon and high-speed trains in these three countries is about 0.1kg~6 kg, 137.9 kg~ 161.5 kg, 99.3 kg~ 765.7 kg and 56.4 kg~ 92 kg. The per capita possession of railway locomotives, railway passenger car, railway wagon and high-speed trains in China is 2.5 kg, 91.6 kg, 635.3 kg and 65.5 kg (Supplementary Table 7), basically close to the level of these three developed countries. Therefore, we can consider the prediction results of this study to be scientific.

7. Comment:

“From the viewpoint of recycling, these four types of products suffer from some difficulties in recycling and a large quantity of treatment.” Such a sentence (and there are many of this kind in the text) is not acceptable for a scientific text. First, no reference is given. Second, “some difficulties” is not specific or descriptive and far too vague. The reader who shall spend time on reading your manuscript deserves more detailed and exact information. Moreover, this sentence should be part of discussion

and not of the results section. Below, you refer to “large weight amounts” and “low recyclability”, but until 2050, recycling capacities can be built up and better recycling technologies be developed, especially in China, where innovation and industry development are very quick!

Response:

Thanks for your comment. We have deleted the sentence and explained it in detail in the discussion. Please check up the tracked-changing manuscript. Thanks !

Remark:

According to the prediction results, WHEE has a large mass and rich resources (Fig.2 and Fig.4). If these resources are recycled, they can reduce energy consumption and pollutant emissions to a certain extent, bring significant economic benefits to the relevant recycling industry, and promote the healthy development of environmental protection and HEE industry. However, compared with other solid waste types such as hazardous waste, WEEE and ELV, HEE recovery is still in its infancy. First of all, due to the large volume, very complex composition and dangerous materials of HEE, HEE disassembly is complicated, and parts are easy to lose. The current level of disassembly is still difficult to meet the needs; Secondly, the number of enterprises with dismantling capacity in China is very small. Taking railway equipment as an example, China established its first train dismantling company in 2021, with a dismantling capacity of 4000 vehicles per year, which is far from meeting the number of trains scrapped; Finally, the scrap recovery management system of HEE has not been improved and there is no corresponding circular economy policy. Therefore, in order to effectively enable anthropogenic circularity, a near-perfect industrial ecosystem should be constructed. Industrial ecology is scaling up a circular economy at the industrial level, with the efficient utilization of materials and product at the core, with the principle of reduction, reuse, remanufacturing, and recycling. This study can provide science-based references for governments and relevant departments to formulate plans and policies concerning anthropogenic circularity and can be of great significance to the sustainable development of the HEE industry.

8. Comment: Title, Figure 2, and text. The indicator “recycling potential” is not defined and not explained. Is it the total mass of all waste equipment or just the fraction that is potentially recyclable and if so, with what technology? This is crucial information and needs to enter the

main text, not just the appendix.

Response:

Thanks. Recycling potential is defined as the recycling amount of generated waste or all the contained materials, which has been well used in many publications*. We add the definition in the revised manuscript.

Remark:

Recycling potential is defined as the recycling amount of generated waste or all the contained materials.^{5, 6} Estimating the generation amount and recycling potential are the most fundamental problems to harness the solid waste.

9. Comment:

Fig. 5 is not clear to me. The red and green shaded areas are not explained (please change color coding to accommodate color-blind readers!), the yellow curve does not show in the pdf version of the manuscript. It looks like that the plot in the pdf is not displayed correctly. Please check and correct.

Response:

Thanks for your comment. In order to make the picture more intuitive, we changed it to error bar chart and changed the color coding. Please check up the tracked-changing manuscript. Thanks!

* Donnini Mancini S, Rodrigues Nogueira A, Akira Kagohara D, Saide Schwartzman JA, de Mattos T. Recycling potential of urban solid waste destined for sanitary landfills: the case of Indaiatuba, SP, Brazil. *Waste Manage Res* **25**, 517-523 (2007).

Graedel TE, Erdmann L. Will metal scarcity impede routine industrial use? *Mrs Bull* **37**, 325-331 (2012).

Zeng X, Gong R, Chen W-Q, Li J. Uncovering the Recycling Potential of "New" WEEE in China. *Environmental Science & Technology* **50**, 1347-1358 (2016).

Ciacci L, Werner TT, Vassura I, Passarini F. Backlighting the European Indium Recycling Potentials. *Journal of Industrial Ecology* **23**, 426-437 (2018).

Xu G, Yano J, Sakai S-i. Recycling Potentials of Precious Metals from End-of-Life Vehicle Parts by Selective Dismantling. *Environmental Science & Technology* **53**, 733-742 (2019).

Islam MT, Huda N. Assessing the recycling potential of "unregulated" e-waste in Australia. *Resources, Conservation and Recycling* **152**, 104526 (2020).

Remark:

Supplementary Fig. 3 Estimated typical materials carbon footprint of China's HEE manufacturing industry in 2000-2050 (a) Fe. (b) Al. (c) Ti. (d) Nd.(e) Total.

10. Comment:

Fig. 6: It is not clear which rate is shown here. Growth rates (%) or flow rates (Mt/yr?)

Response:

Here, the rate is growth rate (%).

11. Comment:

The entire manuscript suffers from a constant mix of introduction, results, and discussion elements. This makes it hard for the reader to follow. For example, in the discussion, suddenly the following phrase appears: "China is the world's largest producer of e-waste 30."

Response:

Thanks for your comment. We deleted this sentence in the text

12. Comment:

The growth rate of e-waste will decrease from 23.7% to 5.6% from 2011 to 2025." That clearly belongs to the introduction. Please make a larger effort to disentangle the different parts of

the manuscript to generate a more logical flow of the text.

Response:

Very good idea. We revise the description of expression to nearly reach the valuable findings.

Remark:

...From 2011 to 2025, the growth rate of e-waste declining from 23.7% to 5.6% indicates a potential substitute of technical product and further implies an ongoing industrial succession.

13. Comment:

The methods section still contains text from the manuscript guideline and the start of the actual content suggests that text by the authors has accidentally been deleted or overwritten.

Response:

We readjusted these sentences and improve the methods. Please check up the tracked-changing manuscript. Thanks !

14. Comment:

Title: Since the case study is on China, this must be reflected in the title. Please change.

Response: okay. The title is changed as *“From wonderland to wasteland: China’s recycling potential of typical high-end equipment and its implications”*.

15. Comment:

Introduction, last paragraph: Since materials are kind of the nutrients of the technical system, it is a good idea to think about ecological succession under the industrial ecology metaphor. However, this has hardly been done in the literature and most readers are not familiar with the concept. Therefore, I think that you must introduce this topic with a language that also non-ecologists understand. In particular, the sentence “In this research, we will also try to examine the succession of technical products from their waste generation.” needs rephrasing. What do you mean by “succession of technical products from their waste generation”?

Response: thanks for your comments. The concept of succession is from the natural ecology. Ecological succession is the process by which natural communities replace (or “succeed”) one another over time. The new field called industrial ecology (learning from nature ecology) is the study of systemic relationships between society, the economy, and the natural

environment. Similarly, technical product is generally replaced by another high-performance product. For instance, CRT-computer has been replaced by LCD-computer or laptop computer. We added some sentences for better understanding.

Remark:

On the other hand, ecology tells that community succession exists while the environment or resource change over a period.²⁵ The growth of one community alters the environment which leads to that community's disappearance and the emergence of a new community which continues to occur in succession. Industrial ecology has the similar phenomenon of succession from emergence, boom, innovation, and substitution of the technical products.²⁶ The new field called industrial ecology (alighting from nature ecology) is the study of systemic relationships between society, the economy, and the natural environment. Similarly, technical product is generally replaced by another high-performance product. Consumer electronics and vehicle industry boomed nearly in the years of 2005 and 2010, respectively. HEE manufacturing, starting from around 2015, is at an advanced stage of industrialization. In this work, we will also try to examine the possible succession of technical products from the insight of their waste generation.

16. Comment:

The results shown in Fig. 1 are redundant, as part C shows the same data as parts B and C, just as different graph. Instead, parts A and B should be placed next to each other, that facilitates visual comparison.

Response:

Thanks for your comment. To make the comparison more intuitive, we redraw the histogram and deleted the redundant pictures. Please check up the tracked-changing manuscript. Thanks!

Remark:

Fig. 1 Comparison of obsolescence amount using the market supply A method and the ownership coefficient method. a obsolescence amount of RL. **b** obsolescence amount of RPC. **c** obsolescence amount of RW. Note: RL: railway locomotives; RPC: railway passenger car ;RW: railway wagon.

17. Comment:

In Fig. 3B, the meaning of the size of the circles is not explained. In the related text you write “From 2010 to 2020, the per capita obsolescence weight increased by 9 kg, which showed that China's transportation facilities were gradually improved and more HEE was put into use with the economic growth, thereby increasing per capita obsolescence quantity too.” This sentence is hard to understand, please rephrase! (“as the economy grew, demand for HEE increased, which led to higher waste flows from this category...”)

Response: Thanks for your comment. The continuous enlargement of the circle represents the continuous increase of the year. We rewrote the sentence, please check up the tracked-changing manuscript. Thanks !

Remark:

In recent years, the total scale of China's comprehensive transport infrastructure has ranked among the top in the world, and the construction of infrastructure such as railway stations and airports has made significant achievements, which has led to the continuous increase in the number of HEEs in China. Under this influence, the generation of WHEE has also increased rapidly, from 0.5Mt in 2000 to 33.2Mt in 2020, and will reach 73.7Mt in 2050, with an annual growth rate of 10.6% (Supplementary Fig. 1g). Among them, railway equipment is the main contributor of WHEE, and the obsolescence mass accounts for more than 99% of the total. In addition, Economic growth significantly accelerates the generation of WHEE²⁴. The regression model between per capita gross domestic product (GDP)

and scrap per capita of HEE is shown in Fig.3. From 2010 to 2020, the scrap per capita increased by 12 kg, which showed that as the economy grew, demand for HEE increased, which led to higher waste flows from this category.

Fig. 3 Relationship between per capita GDP and scrap per capita. Note: The continuous enlargement of the circle represents the continuous increase of the year. Data source from Supplementary Tables 8 and 9.

Reviewer #2

1. Comment:

In the model, it was assumed that the production of HEE increased linearly, and based on this assumption, the futural ownership of HEE over the period 2021-2050 was predicted. Considering the long period, such a simplified linear model may induce significant prediction errors. The authors are suggested to comment or discuss this in more detail on this issue.

Response: Many thanks for your positive comment and constructive suggestion. We changed the model to the logistic model because we thought the model was more scientific, and verified and discussed the output of the model. Please check up the tracked-changing manuscript. Thanks!

Remark:

Data regression. This study uses the time-step method based on data regression to predict the future possession amounts of HEE. The logistic function was put forward by the mathematician Verhulst in the process of studying the problem of population growth. In recent years, the logistic model has been widely used to predict the generation of WEEE and ELV. The growth of the possession amount of product can be divided into four stages: primary introduction, rapid growth and saturation, then final

decline, its growth curve is "S" shape. In addition, China issued a policy in 2021, that is, China will accelerate the construction of a transportation power in the next 30 years, basically build a transportation power by 2035, and fully build a transportation power by 2050. This means that the number of HEE may reach saturation in 2050. Based on the above modeling principles and policies, this study fitted the logistic curve according to the past possession data (Supplementary Table 4 and Supplementary Fig. 5), and the future possession data of HEE can be calculated from Eq. 5:

$$P(t) = \frac{p_{\max}}{1 + ae^{-b(t-t_0)}} \quad (5)$$

where $P(t)$ represents HEE possession amounts in year t ; p_{\max} represents the saturation value of HEE possession amount; and a and b are two parameters describing the growth rate. The simulations were performed by Origin.

As the future generation mass of WHEE is calculated on the basis of the possession amount forecast data, it is necessary to evaluate the possession amount forecast data scientifically. This study will take railway equipment as an example to compare and discuss the railway equipment quantity with that of some developed countries. China has set the goal of developing into a transportation power and a medium developed country by 2050, which means that the possession amount of HEE in China may reach saturation in 2050; At present, railway facilities in some developed countries have been very perfect. Research data show that in recent years, the possession amounts of railway equipment in developed countries such as Japan, Germany and the United Kingdom has changed very little and is basically saturated (Supplementary Table 6). Therefore, this study will compare the predicted per capita possession data of China in 2050 with the per capita possession data of the above three countries in 2019. It is found that the per capita possession of railway locomotives, railway passenger car, railway wagon and high-speed trains in these three countries is about 0.1kg~6 kg, 137.9 kg~ 161.5 kg, 99.3 kg~ 765.7 kg and 56.4 kg~ 92 kg. The per capita possession of railway locomotives, railway passenger car, railway wagon and high-speed trains in China is 2.5 kg, 91.6 kg, 635.3 kg and 65.5 kg (Supplementary Table 7), basically close to the level of these three developed countries. Therefore, we can consider the prediction results of this study to be scientific.

2. Comment: Figure 6 shows the increasing rate of various resources in the next 30 years. It was observed that each kind of resource varies significantly in their increasing rate. It is

widely accepted that the increasing rate decreases gradually. However, Fig. 6 shows that some resources show alternating changes of positive and negative increasing rates. It is an interesting point, but more discussion is required to disclose the mechanism of such variation.

Response: many thanks for your kind comments. We have improved the discussion to imply some important conclusions. Please check up the tracked changing manuscript.

3. **Comment:**

In the section “Analysis of data error and uncertainties,” It is suggested to add a comment on the estimation results concerning the data error and uncertainties. Currently, the authors only described that they had performed the data checking.

Response:

Many thanks for your suggestion. We conducted sensitivity analysis and uncertainty analysis on the results, Please check up the tracked-changing manuscript. Thanks!

Remark:

Sensitivity and Uncertainty Analysis. Due to product substitution, the service life of HEE may change, so sensitivity analysis must be carried out for the change of service life. The study found that when estimating the generation of WHEE in 2010, one year of life extension can reduce the generation of waste by 16.8%, and one year of life reduction can increase the generation of waste by 17.4%. However, there will be no deviation of more than 4.3% in the output after 2015 (Fig.5). This shows that the impact of short-term estimation is significant, but in the long run, the production of WHEE will not be excessively affected by the service life of HEE, which verifies the robustness of the estimation of the production of WHEE in this study.

Based on the collected data (Supplementary Tables 7 and 11), uncertainties of WHEE mass and total stock of Fe and Na in 2050 are performed and presented in Fig5. WHEE mass in 2020 and 2050 in Supplementary Fig. 1g can be excellent reflected by the simulated distribution, show in Fig.5b and Fig.5c. Similarly, resources (like Fe and Nd) stock from Monte Carlo simulation can also cover the forecasting results (Fig.4a) at the maximum probability interval. All the uncertainty analysis can substantially verify and validate the accuracy and

robustness of the above estimation and analysis.

Fig. 5 Sensitivity analysis and uncertainty analysis. **a** Mass of WHEE with 1-year variation. **b** Total mass of WHEE in 2020 affected by each mass. **c** Total mass of WHEE in 2050 affected by each mass. **d** Fe mass of yearly-generated WHEE in 2050 affected by resources content. **e** Nd mass of yearly-generated WHEE in 2050 affected by resources content.

Reviewer #3

1. Comment:

'High-end equipment'. This term is used to describe railway and aviation equipment. It is entirely misleading, and the rationale for using it seems to be the similarity of the acronym to electric and electronic equipment (EEE). EEE is descriptive of the kind, while HEE is not. Many pieces of EEE claim to be high-end (as a simple internet search for 'high-end equipment' shows), but they are not included in the author's definition. Please use a descriptive name.

Response:

Many thanks for your positive comment and constructive suggestion. According to the official

definition, we have classified high-end equipment in detail and added it to the supplementary material.

Remark:

Supplementary Table1 Categories of typical high-end equipment

Typical high-end equipment	Categories
Railway equipment	Railway locomotives (RL)
	Railway passenger car (RPC)
	Railway wagon (RW)
	High-speed trains (HST)
Aviation equipment	Large and medium aircraft (LMA)
	General aviation aircraft (GAA)

2. Comment:

2. RL, RPC, RW, HST etc. Nobody can keep track of all the acronyms that the authors needlessly introduce. They adversely impact the readability of the manuscript. Please write out the names!

Response:

Thank you for your comment. We changed all abbreviations to full names. Please check up the tracked-changing manuscript. Thanks!

3. Comment:

Individual word usage is often problematic. ‘Increasing rate’ is no term. It is rate of increase or growth rate. It needs to be specified in terms of a time step, for example as a rate per year, at least once in particular in a figure. Figure 6 cannot be understood without this information. I do not quite understand how the rates can be so high.

Response:

Thank you for your comment. We take EVs as an example to explain why the product growth rate is high.

Remark:

Due to China's vigorous implementation of the subsidy policy for the purchase of EVs, the improvement of charging piles and the continuous optimization of road driving management environment, the number of electric vehicles has shown a continuous explosive growth trend in recent years. In 2020, the scrap growth rate of EVs has reached 100%.

4. Comment:

I have some problems understanding the referencing. Individual claims or statements are not well

references. For example, the paragraph of the introduction 'Limited researches ..' cites four references after four sentences, but it is not clear which of the prior statements these references refer to. Since methods are cited, I would recommend providing a reference for each method. I also question why none of the references describing the research methods in the introduction are repeated in the methods section.

Response:

Thank you for your comment. We have added references for each method and relevant references in the method section. Please check up the tracked-changing manuscript. Thanks!

Remark:

5. Comment:

Fig. 1A is not referred to in the text. There is no explanation of why two different methods were used, in Fig 1A and Fig 1B, to calculate the same quantities, or how to interpret the differences. Presumably, they provide different empirical basis to estimate a quantity that is not observed and hence constitute some form of triangulation.

Response: Thank you for your comment. We redraw Figure 1 and added an explanation. Please check up the tracked-changing manuscript. Thanks!

Remark:

In this study, the possession coefficient method is used to estimate the generation of WHEE. In order to verify the accuracy of the possession coefficient method, this study uses the market supply A method to verify the accuracy of the possession coefficient method, because the estimated results of the market supply A method are relatively accurate, and there are many application examples. Take railway equipment as an example, the obsolescence amount of RL, RPC and RW in 2020 is estimated to be 759, 1992 and 39539 respectively by using the possession coefficient method; The market supply A method is used to estimate the obsolescence amount of RL, RPC and RW in 2020 as 664, 2180 and 32092, respectively. The research data shows that the average difference between the two methods is 15%, and there is no significant difference (Fig.1). Therefore, the possession coefficient method can accurately estimate the generation of WHEE. The following will use the possession coefficient method to estimate the generation mass of six types of WHEE between 2000 and 2050.

Fig. 1 Comparison of obsolescence amount using the market supply A method and the ownership coefficient method. a obsolescence amount of RL. **b** obsolescence amount of RPC. **c** obsolescence amount of RW .Note: RL: railway locomotives; RPC: railway passenger car ;RW: railway wagon.

6. Comment:

The methods are poorly described. I cannot follow. I do not see that the original method development is appropriately acknowledged. Eq. 1 operates with two different time variables, where one seems to be relative to the other. I have trouble interpreting eq. 4 or understanding what the terms mean and where the data comes from. Please note that cohort or dynamic stock models have been used many years in industrial ecology but have a specific history, which it would be good to acknowledge here.

Response: thanks, we improve the methods for better understanding.

Remark:

7. Comment: Kg measures mass; weight is measured by Newton. (In imperial units, the pound is colloquially used for both, hence the confusion in English.)

Response:

Thank you for your comment. We have replaced the weight in the manuscript with mass. Please check up the tracked-changing manuscript. Thanks!

Remark:

8. Comment:

Displays starting on p. S15 are not legible. Generally, the SI is difficult to read. References are lacking on S2, and so is a description of what is represented in the different parts.

Response:

Thank you for your comment. We have reprocessed the pictures. Because there are too many pictures, we put them in the supplementary materials. We have added relevant references,

Please check up the supplementary material. Thanks!

Reviewers' comments:

Reviewer #1 (Remarks to the Author):

Overview:

The authors prepared a compelling and substantial revision of their original submission. Several of my comments and concerns are addressed well and I am partly satisfied with the authors' response.

There are a number of weaknesses of the current manuscript, however, which severely lower its quality.

I therefore recommend another major revision.

Major comments:

The manuscript needs some restructuring. The link between introduction and results section must come in the introduction, not in the results section. The passage from "WHEE generation. In order to uncover the..." all the way to "are relatively accurate, and there are many application examples" should be rephrased and shorted and moved to the end of the introduction.

The passage: "In recent years, the total scale of China's comprehensive transport infrastructure has ranked among the top in the world,..." is introductory and should be moved to the introduction.

Figure 5 b-e are simple screenshots from a statistical software, but the legend is partly impossible to read. Needs improvement!

Minor comments:

Manuscript now has line numbers, but no page numbers!

Abstract: "72 million tons of steel, 838 tons of aluminium": please check the result on Aluminium: It should be 838 kilotons, right? Also, please make clear in the abstract whether these are cumulative material flows by 2050 or annual flows in 2050!

Reviewer #2 (Remarks to the Author):

The authors presented research work on the prediction of HEE in the future. The topic is of significance for resource recycling and the corresponding policy-making. Generally, the manuscript is well organized. For improving the quality, the following issues may need more clarity of discussion. Specific comments are listed below.

1. In the model, it was assumed that the production of HEE increased linearly, and based on this assumption, the future ownership of HEE over the period 2021-2050 was predicted. Considering the long period, such a simplified linear model may induce significant prediction errors. The authors are suggested to comment or discuss this in more detail on this issue.
2. Figure 6 shows the increasing rate of various resources in the next 30 years. It was observed that each kind of resource varies significantly in their increasing rate. It is widely accepted that the increasing rate decreases gradually. However, Fig. 6 shows that some resources show alternating changes of positive and negative increasing rates. It is an interesting point, but more discussion is required to disclose the mechanism of such variation.
3. In the section "Analysis of data error and uncertainties," It is suggested to add a comment on the estimation results concerning the data error and uncertainties. Currently, the authors only described that they had performed the data checking.

Detailed Response to Reviewers' Comments

Title: From wonderland to wasteland: China's recycling potential of typical high-end equipment and its implications

Manuscript ID: COMMSENG-22-0221A

Reviewer #1

1. Comment:

The authors prepared a compelling and substantial revision of their original submission. Several of my comments and concerns are addressed well and I am partly satisfied with the authors' response.

There are a number of weaknesses of the current manuscript, however, which severely lower its quality.

I therefore recommend another major revision.

Response:

Thank you for your positive comments and the problems pointed out. We have made major revisions to the manuscript.

2. Comment:

The manuscript needs some restructuring. The link between introduction and results section must come in the introduction, not in the results section. The passage from "WHEE generation. In order to uncover the..." all the way to "are relatively accurate, and there are many application examples" should be rephrased and shorted and moved to the end of the introduction.

Response:

Thanks for your comment. We have rephrased and shortened this paragraph and moved it to the introduction section. Please check up the tracked-changing manuscript. Thanks!

Remark:

In order to estimate the generation of WHEE in China, we collected all available data, mainly including the production amount, possession amount, and mass composition of HEE (Supplementary

Tables 3, 4, and 5), Among them, the future possession amount of HEE will be predicted by establishing a logistic model; Then we use the possession coefficient method to estimate the generation of WHEE, and validate the results using the market supply A method. The spatial boundary of this study is Chinese Mainland, and the time boundary is 2000-2050.

3. Comment:

The passage: “In recent years, the total scale of China's comprehensive transport infrastructure has ranked among the top in the world,...” is introductory and should be moved to the introduction.

Response:

Thanks for your advice. We have rephrased this sentence and moved it to the introduction section. Please check up the tracked-changing manuscript. Thanks !

Remark:

With the rapid development of HEE manufacturing and the continuous improvement of transport infrastructure, an increasing number of resources are flowing into this new industry.

4. Comment:

Figure 5 b-e are simple screenshots from a statistical software, but the legend is partly impossible to read. Needs improvement!

Response:

Thanks for your comment. We have added a legend to the figure. Please check up the tracked-changing manuscript. Thanks.

Remark:

Fig. 5 Sensitivity and uncertainty analysis. a Mass of WHEE with 1-year variation. **b** Total mass of WHEE in 2020 affected by each mass. **c** Total mass of WHEE in 2050 affected by each mass. **d** Fe mass of yearly-generated WHEE in 2050 affected by resources content. **e** Nd mass of yearly-generated WHEE in 2050 affected by resources content.

5. Comment:

Manuscript now has line numbers, but no page numbers!

Response:

Thanks for your advice. We have added page numbers to the manuscript. Please check up the tracked-changing manuscript. Thanks !

Remark:

The obtained results show that their total recycling potential has experienced a rapid growth, the cumulative obsolescence mass in 2020 exceeded 33 million tons, and it is expected to reach another 74 million tons by 2050., roughly twice the amount in 2020. By 2050, waste high-end equipment in China will contain at least 72 million tons of steel, 838 kilotons of aluminum, 2,539 tons of titanium and 223 tons of neodymium (cumulative material flows).

6. Comment:

Abstract: “72 million tons of steel, 838 tons of aluminium”: please check the result on Aluminium: It should be 838 kilotons, right ? Also, please make clear in the abstract whether these are cumulative material flows by 2050 or annual flows in 2050!

Response:

Thank you for your careful discovery. We have made the modifications and checked all the data in the manuscript. We have indicated the cumulative material flows in the manuscript. Please check up the tracked-changing manuscript. Thanks !

Remark:

The obtained results show that their total recycling potential has experienced a rapid growth, the cumulative obsolescence mass in 2020 exceeded 33 million tons, and it is expected to reach another 74 million tons by 2050., roughly twice the amount in 2020. By 2050, waste high-end equipment in China will contain at least 72 million tons of steel, 838 kilotons of aluminum, 2,539 tons of titanium and 223 tons of neodymium (cumulative material flows).

Reviewer #2

1. Comment:

The authors presented research work on the prediction of HEE in the future. The topic is of significance for resource recycling and the corresponding policy-making. Generally, the manuscript is well organized. For improving the quality, the following issues may need more clarity of discussion. Specific comments are listed below.

1.

In the model, it was assumed that the production of HEE increased linearly, and based on this assumption, the futural ownership of HEE over the period 2021-2050 was predicted. Considering the long period, such a simplified linear model may induce significant prediction errors. The authors are suggested to comment or discuss this in more detail on this issue.

2.

Figure 6 shows the increasing rate of various resources in the next 30 years. It was observed that each kind of resource varies significantly in their increasing rate. It is widely accepted that the increasing rate decreases gradually. However, Fig. 6 shows that some resources show alternating changes of positive and negative increasing rates. It is an interesting point, but more discussion is required to disclose the mechanism of such variation.

3.

In the section "Analysis of data error and uncertainties," It is suggested to add a comment o

n the estimation results concerning the data error and uncertainties. Currently, the authors only described that they had performed the data checking.

Response: Many thanks for your positive comment and constructive suggestion. Since your review opinion this time is *the same as the first sound review opinion*, we have made modifications and replied to the previous review opinion. In this time, we have improved again to add the additional discussion. Please review the results of our last modification. Thanks!

REVIEWERS' COMMENTS:

Reviewer #1 (Remarks to the Author):

The authors prepared a second compelling and substantial revision of their first revision.

The manuscript has improved substantially in quality, and all my comments are satisfactorily addressed except for one, which is why another minor revision is needed.

Remaining issue: The distinction between annual flows and cumulative flows is still not clear.

Here, I cite examples from the abstract, but the entire manuscript must be checked and revised for clarity!

(1) "The cumulative obsolescence mass in 2020 exceeded 33 million tons, ..."

 the term "cumulative obsolescence mass" is not defined in our field, you probably mean the "cumulative waste material between 2000 and 2020". Whenever indicating cumulative quantities, you must state the time period over which the flows are accumulated/summed up!

This is crucial especially when cumulative and annual quantities are reported side by side, as it is the case here!

(2) "... expected to reach another 74 million tons by 2050, roughly twice the amount in 2020."

 Is it annual (then please state so!) or cumulative?

So, is it "another 74 Mt between 2020 and 2050, roughly twice the amount between 2000 and 2020", or "another 74 in 2050, roughly twice the amount in 2020."? That is not clear! Your phrase mixes 'in' and 'by', so annual and cumulative quantities.

(3) "waste high-end equipment in China will contain at least 72 million tons of steel, 838 kilotons of 19 aluminum, 2,539 tons of titanium and 223 tons of neodymium (cumulative material flows)"  Again, please state the time frame (2020-2050?) over which the cumulative material flow was calculated!

Reviewer #2 (Remarks to the Author):

The authors have addressed/clarified issues raised. The manuscript is ready for publication.

Detailed Response to Reviewers' Comments

Title: From wonderland to wasteland: China's recycling potential of typical high-end equipment and its implications

Manuscript ID: COMMS-22-0221B

Reviewer #1

1. Comment: The authors prepared a second compelling and substantial revision of their first revision. The manuscript has improved substantially in quality, and all my comments are satisfactorily address except for one, which is why another minor revision is needed. Remaining issue: The distinction between annual flows and cumulative flows is still not clear. Here, I cite examples from the abstract, but the entire manuscript must be checked and revised for clarity!

(1) "The cumulative obsolescence mass in 2020 exceeded 33 million tons, ..."  the term "cumulative obsolescence mass" is not defined in our field, you probably mean 3 the "cumulative waste material between 2000 and 2020". Whenever indicating cumulative quantities, you must state the time period over which the flows are accumulated/summed up! This is crucial especially when cumulative and annual quantities are reported side by side, as it is the case here!

(2) "... expected to reach another 74 million tons by 2050, roughly twice the amount in 2020." -> Is it annual (then please state so!) or cumulative? So, is it "another 74 Mt between 2020 and 2050, roughly twice the amount between 2000 and 2020", or "another 74 in 2050, roughly twice the amount in 2020."? That is not clear! Your phrase mixes 'in' and 'by', so annual and cumulative quantities.

(3) "waste high-end equipment in China will contain at least 72 million tons of steel, 838 kilotons of 19 aluminum, 2,539 tons of titanium and 223 tons of neodymium (cumulative material flows)"  Again, please state the time frame (2020-2050?) over which the cumulative material flow was calculated!

Response: very good questions and reminding. Here, cumulative is total!! It is the total amount of all typical high-tech equipment in one year as annual data. The three points you gave are similar. We have revised it through the paper. Much appreciate!